# Factors Associated with Dietary Diversity in Community-Dwelling Brazilian Older Adults

**DOI:** 10.3390/foods13213449

**Published:** 2024-10-29

**Authors:** Ligiana Pires Corona, Graziele Maria Silva, Carolina Neves Freiria

**Affiliations:** 1School of Applied Sciences, University of Campinas, R. Pedro Zaccaria, 1300, Limeira 13484-350, Brazil; nutrigrasilva@gmail.com; 2School of Nutrition, Universidade São Francisco, Bragança Paulista 12916-900, Brazil; carolnfreiria@gmail.com

**Keywords:** dietary diversity, diet quality, food groups, cross-sectional study, older adults

## Abstract

Background: Older populations are at high risk of nutritional inadequacy and monotonous diets, and assessing dietary diversity can be a practical measure to indicate groups at nutritional risk. We aimed to explore the dietary diversity of older adults enrolled in primary health care services in Brazil and to evaluate its associated factors. Methodology: In this cross-sectional study, we evaluated the dietary diversity score (DDS) of 581 participants (≥60 years) registered in primary care services. All foods mentioned in a 24 h food recall were classified into 10 groups, and factors associated with the DDS were analyzed using hierarchical linear regression models in two blocks: (1) sociodemographic and (2) health conditions and lifestyle. Results: The mean DDS was 5.07 (±1.34), and 67.5% of the sample reached the minimum dietary diversity (≥5 groups). In the final model, income, previous diagnosis of cancer, and sporadic intake of alcohol were positively associated with DDS. In contrast, cognitive decline, sedentary lifestyle, and anorexia of aging were negatively associated with DDS. Conclusions: These findings show that the entire structural, economic, and social system needs to facilitate access to quality food, adequate places and conditions for the practice of physical activity, and policies regarding tobacco and alcohol abuse, in addition to nutritional guidance.

## 1. Introduction

The phrase “eat a variety of foods” is a public health recommendation first introduced in the early 20th century in response to the prevalence of nutritional inadequacies, based on the premise that consuming a wide variety of foods will ensure an adequate intake of essential nutrients and lead to better diet quality and better health outcomes [1]. Even so, a lack of dietary diversity is still a serious problem, particularly in the developing world because their diets are predominantly based on starchy foods and often include few or no animal products and few fresh fruits and vegetables [2,3].

The literature has consistently shown that the usual diet of older adults is monotonous, and this is a worrisome scenario because population aging has been occurring at an accelerated rate all over the world. According to the World Health Organization (WHO), in 2020, the global population aged 60 years and over was around over 1 billion people, representing 13.5% of the world’s population, with two-thirds of older people living in low- and middle-income countries (LMICs) [4]. During the United Nations Decade of Healthy Ageing (2021–2030), it is expected that the number of older adults will increase from 1.1 to 1.4 billion by 2030 worldwide [5].

In aging, many factors can cause a worsening in the dietary pattern. In addition to the accumulation of noncommunicable diseases (NCDs), other factors such as psychological factors, cognitive decline, lower purchasing power, lack of social support, isolation, and environmental or functional difficulty in accessing food may further contribute to this population presenting a monotonous diet with low nutritional quality and that is poor in fiber, vitamins, and minerals [6,7]. Malnutrition in older adults is associated with prolonged hospitalization, increased risk of falls, decreased physical function, poorer quality of life, sarcopenia, frailty, and increased mortality [6,8].

Some studies have already shown that low dietary diversity can be a good indicator of nutritional inadequacy in older adults. For example, a study carried out with elderly Iranian women found a significant positive correlation between dietary diversity and indicators such as the Mean Adequacy Ratio (MAR) and Nutrient Adequacy Ratio (NAR) of all 12 nutrients studied; therefore, it was an appropriate proxy for the probability of nutritional adequacy [9]. Another study carried out in South Africa observed that this indicator had a high capacity to identify respondents with an inadequate diet but less capacity to identify respondents with a nutritionally adequate diet; therefore, it can be a simple tool to identify elderly people at risk of food and nutritional insecurity [10].

The Minimum Dietary Diversity for Women (MDD-W), developed by the Food and Agricultural Organization (FAO) of the United Nations in 2016, is a proposal for a single indicator to assess dietary quality in women of reproductive age. It is based on the premise that people who achieve minimal diet diversity, i.e., consuming five or more food groups, are expected to have a higher likelihood of meeting their micronutrient intake recommendations compared to those who consume fewer food groups [11,12,13].

Despite being proposed for younger women, the MDD-W has been used as a proxy measurement for diet quality and micronutrient adequacy, mainly because it is easy to use, showing important associations with nutrient adequacy. The ten groups are mutually exclusive and include sources of all nutrients that are important to health, such as energy, vitamins, animal and vegetal proteins, and minerals. In addition, fruits and vegetables are divided into those that are either rich in vitamin A or not, another advantage of this indicator. As the groups are quite comprehensive, including food items from the most diverse cultures, this tool can be used to compare populations from different countries and continents [11].

Because of all these advantages, the MDD-W has been used in LMICs, including Latin American countries, in several other populational groups, such as other age groups and men [13,14]; however, very few studies have used this tool in older populations. A study by Rodríguez-Ramírez and collaborators (2022) evaluated data from more than 10,000 Mexicans of both sexes and all age groups (≥1 year old) using this indicator and described a mean DDS of 4.75 for older adults, and about 68.4% of them did not reach the minimum dietary diversity (MDD), considering, however, that adults and older populations should have a minimum consumption of six groups to achieve a greater probability of adequacy [13]. A study conducted with older adults in Thailand found a mean DDS of 18.4 using an adaptation of the version adopted by the FAO, which ranges from 0 to 32 points, which means that, on average, the score was slightly higher than 16, the value that would correspond to half the score in this version [15].

As dietary diversity is a key element of diet quality, DDSs can be translated into good indicators for use in public health, as they are relatively simple to collect, that is, they are applicable to field research, they have easy-to-calculate scores, and their analysis and interpretation are quite direct, that is, they are without subjectivity, especially in developing countries with a rapid aging process, such as Latin American locations. Therefore, this study aims to explore the dietary diversity of older adults enrolled in primary health care services in Brazil and to evaluate its associated factors.

## 2. Methods

### 2.1. Design, Sample, and Data Collection

This was a cross-sectional study that originally aimed to evaluate data on food consumption and micronutrient deficiency in older adults in three cities in the State of Sao Paulo, Brazil (Campinas, Limeira, and Piracicaba, where there is a campus of Universidade Estadual de Campinas). The study sample was estimated considering the total number of inhabitants aged 60 years and older using official data of population estimates for 2018 and considering a prevalence of 60% of older adults with a deficiency in at least one of the evaluated nutrients, a sampling error of 10%, and a 95% confidence level. Then, the minimal sample was estimated at 600 older adults [16].

The Health Departments of each municipality indicated some public primary care units around the city (which had space and time available for data collection); thus, the sample was not probabilistic, and it was conveniently decided, with the only recommendation that it represent all the administrative areas of the municipality. At each health unit, users who met the inclusion criteria (aged 60 years or older, living in one of the participating municipalities, being registered in the Family Health Strategy (FHS)) were invited by doctors, nurses, or other health professionals at the health unit to participate in the research; again, there was no probabilistic sampling [16,17].

This study interviewed 611 older adults in the three participating cities (Limeira: n = 176; Piracicaba: n = 187; Campinas: n = 248) from October 2018 to December 2019, but 30 observations were excluded for not having complete questionnaire data or for presenting implausible caloric values (<500 or >5000 kcal). Thus, the final sample that was considered in the study of food consumption data was 581 participants. The research protocol was approved by the Research Ethics Committee of the University of Campinas under protocol number 95607018.8.0000.5404, and all evaluations were performed only after the participants signed a confirmation consent term.

### 2.2. Study Variables

For the dietetic evaluation, a 24 h food recall (24 h-FR) was applied by previously trained interviewers, adopting some strategies to minimize possible biases, such as using photographic guides to help estimate portions, and the interview was carried out following the MPM (Multiple-Pass Method) consisting of 3 sequential steps as previously described: (1) initial report by the participant in chronological order without interruption; (2) details of food, drinks, meals, and times reported; and (3) chronological review of foods reported on each occasion and questions about additional consumption beyond what was reported, frequently forgotten items, etc. No data collection was carried out on Mondays to avoid capturing Sunday consumption, which tends to be atypical.

For the construction of the dietary diversity score (DDS), all foods mentioned in the 24 h-FR were classified into the following groups: (1) starches (grains, white roots and tubers, and plantains); (2) pulses (beans, peas, and lentils); (3) nuts and seeds; (4) dairy products; (5) meat, poultry, and fish; (6) eggs; (7) dark green leafy vegetables; (8) other vitamin-A-rich fruits and vegetables; (9) other vegetables; and (10) other fruits. One point was assigned for the consumption of at least 15 g of each food group, and zero points were assigned if there was no consumption or less than 15 g. A total of ten points could be obtained for the maximum-variability diet, and higher scores indicated greater dietary diversity as more food groups were consumed. Adequate minimum dietary diversity (MDD) was achieved by participants with a score ≥ 5 [11].

The survey questionnaire contained questions about personal data, sociodemographic conditions, health, and lifestyle, in addition to physical measurements (Appendix A). For the present analysis, we selected the main sociodemographic characteristics, such as gender (male; female), age group (60 to 69 years old; 70 years old or more), ethnicity (Caucasians; African American; and others), living arrangement (living with other people; living alone), schooling (from 0 to 4 years of study; 5 or more years of study), total family income per capita in current minimum wages (up to 1 minimum wage; from 1 to 2 minimum wages; 2 or more minimum wages). The value of the minimum wage was considered the one in effect at the time of the interview (BRL 954.00 for participants interviewed in 2018 and BRL 998.00 for respondents in 2019).

Social support was measured using the Brazilian-adapted and validated version of the Medical Outcomes Study Social Support Survey (MOS) [18]. The instrument contains 19 questions, assessing 5 dimensions of social support: material, affective, emotional, positive social interaction, and information, and the total score varies from 0 to 100 points, where higher scores indicate better-received support. No cut-off point is proposed for the scale.

The self-reported presence of NCDs was assessed by asking whether the volunteer had received any of the following diagnoses from a physician or other health professional: hypertension, diabetes, cardiovascular disease, tumor or cancer, stroke, chronic lung disease (asthma, emphysema, etc.), arthritis/arthrosis or rheumatism, osteoporosis, allergy, and kidney diseases. For the present study, the most frequent conditions or those with the greatest association with possible dietary changes were used: hypertension, diabetes, cardiovascular diseases, and cancer.

Anorexia of aging was assessed using the Simplified Nutritional Appetite Questionnaire (SNAQ), an instrument with 4 simple questions about appetite, validated in Brazil [19]. The score can vary from 4 (worst) to 20 (best), and anorexia is identified when results are ≤14 points.

Cognitive status was assessed using the Cognitive Skills Screening Instrument—short version (CASI-S), validated in Brazil, indicating the presence of cognitive decline if the score is less than 23 in individuals aged 60 to 69 years and less than 20 for those aged 70 years or older [20].

For the physical activity practice, we asked about the frequency and duration of any leisure activity, and those who reported 150 min or more of moderate activity per week were classified as active; those who reported some practice, but did not complete a minimum of 150 min of activities in the week were classified as insufficiently active; and those who did not practice leisure-time physical activity were classified as sedentary, according to WHO cut-offs [21]. Regarding smoking, for the present analysis, participants who reported current smoking were considered smokers (former smokers were grouped in the non-smoker category). The report of alcohol consumption was low in our sample; therefore, it was classified as never/rarely; 1 to 4 times a month; at least 1 time a week.

### 2.3. Statistical Approach

First, exploratory analyses of the data were performed, describing the distributions of continuous and frequencies of categorical variables. Adherence to normal distribution was verified using the Shapiro–Wilk test, and as it was confirmed, the differences between the groups were evaluated using either Student’s *t*-test for dichotomous explanatory variables or simple linear regression for variables with more than 2 categories. Differences between the explanatory variables and the prevalence of the MDD were assessed using the x^2^ test.

The study hypothesis is that many factors are associated with lower dietary diversity in the sample, including health conditions and lifestyle indicators, which are also influenced by social determinants. Therefore, we opted to assess the factors associated with the DDS using hierarchical linear regression analyses, where the variables that were significant in the bivariate analysis were inserted in 2 hierarchical blocks in the multiple models: (1) sociodemographic: sex, age group, ethnicity, living arrangement, marital status, per capita family income, education, and score of social support; and (2) health conditions and lifestyle indicators: self-reported NCDs, anorexia of aging, cognitive decline, physical activity, smoking, and alcohol intake. Variables that remained significant or that adjusted other variables were maintained in the final model. All analyses of the study were performed using the STATA^®^ software, version 14, with a critical level of *p* < 0.05.

## 3. Results

Among the participants, 69.4% were female, the mean age was 69.5 years old, 16.9% lived alone, most of them had low income (74.9% had an income lower than two minimum wages), and 51.2% had less than 5 years of formal education. The most prevalent condition was hypertension (61.5%), 18.8% had cognitive decline, and 25.6% presented with anorexia of aging. In addition, most of them were not physically active (54.9%).

The most consumed food groups among the 581 participants in the study were starches (100% intake), followed by meat, fruits, and dairy products. Table 1 shows the prevalence of consumption by food groups according to sex and age group. Women consumed significantly less legumes, and more dark green leafy vegetables, other fruits and vegetables rich in vitamin A, and other fruits. Regarding the age group, older participants consumed significantly more dairy products and fewer eggs.

The mean dietary diversity (DDS) for the total sample was 5.07 (SD = 1.34), with 392 participants (67.5%) reaching the minimum dietary diversity (MDD). Table 2 shows the sociodemographic characteristics of the sample concerning dietary diversity. We observed that the higher the per capita family income, the higher the mean DDS, but the prevalence of the MDD was similar in the two higher-income categories. Schooling over 5 years was also associated with a higher DDS and a higher prevalence of the MDD. There was no difference in the mean value of the DDS and the prevalence of the MDD regarding the other variables. The mean social support score was 76.18 points for participants with the MDD and 83.79 for those who did not reach the MDD (*p* < 0.001).

Table 3 displays the dietary diversity according to the selected health conditions and lifestyle characteristics. Previous diagnosis of cancer was associated with a higher DDS and a higher prevalence of the MDD, while the presence of cognitive decline showed an inverse association, i.e., a lower DDS and a lower prevalence of the MDD. Both the DDS scores and the MDD prevalence were associated with anorexia of aging. The other NCDs showed no statistically significant difference.

Regarding lifestyle indicators, sedentary participants and smokers had significantly lower averages in terms of the DDS and a lower prevalence of the MDD. The DDS was also significantly higher in older adults who consumed alcoholic beverages sporadically (1 to 4 times a month).

Table 4 shows the results of the hierarchical multiple regression analysis of the factors associated with the DDS. In the final model, income, previous diagnosis of cancer, and sporadic intake of alcoholic beverages remained positively associated with the DDS. In contrast, the presence of cognitive decline, a sedentary lifestyle, and the presence of anorexia of aging were negatively associated with the DDS.

## 4. Discussion

### 4.1. Prevalence of Consumption by Food Groups

In our study, women consumed significantly less legumes and consumed more dark green leafy vegetables, other fruits and vegetables rich in vitamin A, and other fruits. This same trend of a difference between the sexes was observed in Brazilian national studies with adults. For example, according to the Surveillance System for Risk and Protective Factors for Chronic Diseases by Telephone Survey (Vigitel), in a set of 27 cities surveyed in 2021 (capitals of the 26 Brazilian states and the Federal District), the frequency of consumption of beans on five or more days of the week was higher among men (65.9%) than women (55.8%), and the frequency of consumption of five or more daily servings of fruits and vegetables was higher among women (26.4%) than men (16.9%) [22].

The literature usually discusses that men are less concerned with their health conditions, and it is also assumed that they have less knowledge about current dietary recommendations, considering that the consumption of vegetables may be less important for health [23]. A study with English older adults also described that gender differences in terms of fruit and vegetable intake were substantially attenuated by controlling for nutritional knowledge, that is, the lower nutritional knowledge of men explains a significant part of their lower consumption [24].

We also observed that older individuals had a higher consumption of dairy products and a lower consumption of eggs. The Brazilian National Dietary Survey, conducted in a sub-sample of the Household Budget Survey (NDS-HBS, in Portuguese Inquérito Nacional de Alimentação-Pesquisa de Orçamento Familiar—INA/POF) already showed that older persons had a higher prevalence of consumption of whole milk and a lower consumption of eggs compared to adults [25], but studies that compare the intake of specific food groups between different age groups in older groups are rare. A study that compared the dietary intake of two cohorts of participants in the SABE Study, in Sao Paulo, showed that those born between 1946 and 1950 (younger cohort) had a significantly lower consumption of dairy products than the older cohort (born between 1936 and 1940); when the prevalence was evaluated separately by sex, this difference was significant only among men [26]. It is not possible to know the reasons involved in these differences, but it is possible to raise the hypothesis that the older participants in the present study had already received guidance from health professionals about the importance of dairy products in bone health, since they are the main sources of dietary calcium. Previous studies have already described that older adults are more likely to make positive dietary changes due to a greater understanding of the benefits of a proper diet or due to chronic diseases requiring the adoption of healthier habits [27].

A study that analyzed the trends in dietary intake among older Americans from 1977–2010 showed that milk was the third major source of energy among older Americans in 1989–1991 and the fourth in NHANES in 2005–2010, showing that maybe younger cohorts are indeed decreasing their dairy consumption; on the other hand, dairy desserts were in sixteenth in the food rank and jumped to fifth place in the same period [27]. This may also reinforce a shift from unprocessed or minimally processed food to ultra-processed products in older adults.

Considering the differences described here in the group in terms of the consumption of eggs, no consistent data were found to compare our results. In the relatively recent past, a restrictive consumption of eggs has been recommended due to their cholesterol content, mainly in people with dyslipidemia or at risk of cardiovascular diseases; however, this recommendation is outdated and no longer used in dietary guides in most countries, including Brazil, since the literature has already shown that moderate egg consumption has little impact on the lipid profile [28]. However, it is possible to hypothesize that older participants continue to restrict their egg consumption considering these older recommendations, but it is not possible to discuss this issue in depth based on the design and variables of the present study.

### 4.2. Factors Associated with Dietary Diversity

Regarding the mean DDS and prevalence of the MDD described here, it was observed that the average dietary diversity score was five food groups, and about two-thirds of the older adults reached the MMD (consumption of at least five of the ten food groups).

As already presented in the introduction, only a few studies have analyzed dietary diversity in older populations using various indicators, which makes it difficult to compare these data. Recently, a study by Rodríguez-Ramírez and collaborators (2022) described a mean DDS of 4.75 for older adults in Mexico, lower than that found in the present study; the prevalence of not reaching the MDD was 68.4% [9], agreeing with the results found in our sample, which could indicate that Latin American older populations may have similar conditions regarding diversity. Indeed, both countries share some characteristics in common, such as both being classified as upper-middle-income economies by the World Bank Atlas method [29]; both have an ongoing nutritional transition, with a double burden of malnutrition (nutritional deficiencies along with a higher prevalence of obesity); and they also found dietetic habits based in starchy staples, beans, dairy foods, and fruits [9], which are also traditional habits from other Latin American countries.

Regarding the factors associated with the DDS score, the present study found that a higher diversity was positively associated with income, previous diagnosis of cancer, and sporadic alcohol intake; on the other hand, the presence of cognitive decline, sedentary lifestyle, and anorexia of aging were associated with lower diversity.

A population study in Spain found that the DDS was significantly higher in women, non-smokers, and those with lower educational levels [30]. In the present study, there was no difference regarding sex, and a higher income was associated with higher diversity. Some studies have already shown that a higher income and education are associated with better diet diversity, both in older adults and in other population groups [14,31,32,33]; however, Otsuka et al. (2017) did not observe an association between diversity and education level in the Japanese population [34], in line with the result presented here, because despite schooling being associated with the DDS in the crude analyses, it did not remain significant in the adjusted model. Possibly, this occurred because both variables are important indicators of socioeconomic status and are closely related in developing countries, and when placed together in a regression model, they can attenuate each other’s effect.

The relationship between socioeconomic status and the quality of the diet is still controversial in the literature. In general, it is considered that more educated individuals possibly value healthy eating more, since they understand the importance of the prevention or control of NCDs, also tending to have healthier behaviors [35,36]. However, a higher income and education may also be associated with greater access to lower-quality foods, such as ultra-processed foods [37,38]. In the case of older populations, income possibly has a greater effect than education in obtaining higher dietary diversity; moreover, the degree of processing was not analyzed in the diversity indicator, so it is possible that this effect exists and was not captured in our results.

In our study, social support was not significant in the final model, despite being widely described in the literature as an important factor associated with better dietary quality in older people and a lower likelihood of food insecurity [39,40,41,42,43]. A qualitative study that analyzed food choices and access to food among low-income older adults pointed out that social interactions, especially with family and friends, can positively affect eating behavior, increasing the consumption of healthier meals and decreasing the consumption of fast meals, such as toast and cereal [44]. Possibly, in the present study, social support lost significance because it was also associated with several other model variables, such as income and physical activity.

Reporting a previous diagnosis of cancer, on the contrary, was associated with a higher score on the EDD. These results could be explained by the hypothesis that changes in eating habits and lifestyle may occur after the diagnosis of the disease, and as our study was cross-sectional, these changes may have already occurred in the participants with a previous diagnosis. In 2022, the American Cancer Society published a new guideline on nutrition and physical activity for cancer survivors, emphasizing the importance of lifestyle changes from the moment of diagnosis for a better tolerance to treatment, the prevention of recurrence, and the possible delay in mortality [45]. Liu and colleagues (2021) conducted a cohort study of nearly 18,000 elderly people to assess the impact of changes in dietary diversity on mortality in China and found that those who had a drastic improvement in DDS throughout follow-up also had an increase in mortality, and the authors report that this group had more underlying diseases, which may explain this change in the DDS [46].

Anorexia of aging showed a consistent negative association in the results presented here. Anorexia of aging is known as a multicausal syndrome related to diseases, polypharmacy, and physiological factors intrinsic to aging itself, in addition to social, psychological, environmental, and lifestyle factors that can further affect eating habits and nutritional status, leading to weight loss and malnutrition [47]. A previous study of our research group showed that the intake of most nutrients is significantly lower in older adults with anorexia, except for carbohydrates, which may point to poorer-quality diets of these persons, preferring palatable and easy-to-eat items [48]. Anorexia screening is essential to prevent malnutrition, sarcopenia, frailty, cognitive decline, and mortality in older adults and should be considered in any geriatric assessment [8,49].

We also found a significant negative association between the DDS and cognitive decline, which agrees with other results already described in the literature, such as cross-sectional studies in China and Taiwan [50,51]. Longitudinal studies have also shown that less dietary diversity is associated with a higher incidence of cognitive decline over time [34,52,53,54], but almost all these studies were conducted in eastern countries, such as China, Japan, and Taiwan, so it is not known whether diversity also has an impact on cognition in other populations.

However, in the results presented here, it is not possible to identify whether cognitive decline would be a determinant or a consequence of lower dietary diversity, since both directions of association would be plausible. In addition to the fact that less dietary diversity would be responsible for a lack of several essential nutrients for mental health, older adults who already present cognitive deficits may have difficulty in various activities of daily living related to food intake, such as shopping and preparation of items, difficulty in bringing food to the mouth, difficulty swallowing, among others, which can reduce food variety and compromise nutritional status [6].

Lifestyle characteristics also showed an important association with dietary diversity in the present study; sedentary individuals had lower DDS and MDD prevalence than active ones; those who consumed alcoholic beverages sporadically (1 to 4 times a month) also had higher diversity than those who did not drink or had weekly consumption; and smokers also had less diversity in the crude analyses, but this association lost significance in the adjusted model. These results were, in part, expected, as it is described in the literature that individuals with better lifestyle practices generally also have better diet quality [35,55,56].

Few studies have evaluated the association of diet diversity as an indicator of nutrition with other lifestyle markers, and the results so far seem to indicate that smoking is a factor more strongly associated with lower diversity (although this variable did not remain significant in our final model); the results described regarding alcohol consumption and physical activity were null or conflicting. A study with older Japanese people found a negative association between diet diversity and smoking and alcohol consumption (classified as current, previous, or never, without considering frequency) but not with physical activity, in addition to another study in Spain, which also described a negative association with smoking and a non-significant association with the level of physical activity; alcohol consumption was not analyzed [30,33]. Another Japanese study also described a negative association with smoking and a non-significant association with alcohol consumption (in volume/day) and did not assess physical activity [34].

The positive association found in the present study between dietary diversity and sporadic alcohol intake can be explained by the various existing recommendations that relate moderate consumption with lower risks of diseases and cardiovascular mortality [57]. These recommendations are generally based on Mediterranean-style diets, in which moderate wine consumption is one of the most marked characteristics [58]. It has also been related to better cognitive performance in older adults [59]. Thus, it is possible that older adults who pay more attention to their diet also consider the recommendation to associate a varied diet with moderate alcohol consumption.

However, it is important to mention that more recent studies have shown that perhaps there is no safe alcohol consumption limit, especially for older adults. A recently published study performed a combined analysis of data from nearly 600,000 participants in 83 prospective studies and discussed the complex and diverse potential mechanisms by which alcohol consumption can exert cardiovascular effects, highlighting that higher alcohol consumption was approximately linearly associated with increased risk of all stroke subtypes, coronary heart disease excluding myocardial infarction, heart failure, and several less common subtypes of cardiovascular disease. The authors discussed that although the threshold for the lowest risk of all-cause mortality was around 100 g per week, for cardiovascular disease subtypes other than myocardial infarction, there were no clear thresholds; therefore, their data support the adoption of minimum limits of alcohol consumption, considerably below what is recommended in most current guidelines [57].

### 4.3. Strengths and Limitations

This study has limitations that must be considered when interpreting its results. First, the method used to assess food consumption (24 h-FR) is subject to memory bias, which may be compromised among older adults, although all interviewers had been trained and used photographic manuals to minimize possible errors. It is also important to mention that the 24 h-FR is a method that describes the interviewee’s current food consumption and may not be suitable for estimating usual consumption. However, we believe that this bias was minimized considering that: (1) this is the method used by the indicator according to the FAO (2016) [11]; (2) older adults tend to have a monotonous diet most of the time, varying little between days [7]; and (3) it is well described that if the sample is large enough, such as the one studied here, even a single day of consumption per individual can be used to estimate the average habitual consumption of the population, eliminating extreme values due to population distribution, and, to estimate intra-individual daily variation, it is usually statistically more efficient to increase the number of individuals in the sample than to increase the number of days above 2 days per individual [17,60].

Moreover, some important variables that could be related to a higher dietary diversity were not available in the questionnaire and therefore were not analyzed, such as access to markets and places where food is purchased, food environment, or functional status. It is possible that if they were included, they would have brought different results to the study. However, the sample had good functionality, considering that all the respondents went to the research site (primary care units) and considering the high prevalence of physical activity.

Furthermore, the diversity indicator used here (MDD-W) has not been validated in older populations. However, it has been the method indicated by the FAO to assess dietary diversity in different populations, not only in women of reproductive age, and it has been used with other groups, such as children and adults, including Latin Americans [11]. Rodríguez-Ramírez and collaborators (2022), for example, used this indicator to analyze data from more than 10,000 Mexicans of both sexes and all age groups (≥1 year of age) [13]. The Estudio Latino Americano de Nutrición y Salud (Latin American Study of Nutrition and Health—ELANS), conducted in eight countries in South America (Argentina, Brazil, Chile, Colombia, Costa Rica, Ecuador, Peru, and Venezuela), also used this indicator to assess the dietary diversity of individuals aged between 15 and 65 years of both sexes [14].

Finally, it is important to discuss that LMICs, such as Brazil, are undergoing an accelerated and heterogeneous nutritional transition, characterized by an increased intake of unhealthy fats, refined carbohydrates, and added sugar, i.e., the use of dietary diversity scores can be a major limitation in these contexts, as they do not capture the three important dimensions of diet quality (adequacy, variety, and moderation); in particular, the moderation dimension is often absent or inadequately evaluated [61,62]. However, its objective is to evaluate the minimum presence of the food group, being considered as a reliable proxy of adequacy, in addition to diversity.

## 5. Conclusions

The results of this study showed that among older adults using primary care services in three Brazilian cities, the diet diversity was higher in those with better socioeconomic conditions, with previous diagnoses of cancer, and with preserved appetite and cognition. Also, people who practiced physical activity and consumed alcoholic beverages 1 to 4 times a month had higher diversity. These findings advance the literature because they highlight the need for behavioral interventions to target multiple behaviors and for health prevention efforts, considering the interactive nature of these behaviors, as well as the need to provide adequate financial support to guarantee the minimum diversity for older adults. It is not possible to attribute these health behaviors and lower dietary diversity to individual causes; rather, it is the entire structural, economic, and social system that needs to facilitate access to quality food, adequate places, and conditions for the practice of physical activity, as well as policies regarding tobacco and alcohol abuse, in addition to nutritional guidance.

The practical implications of our results are that the use of dietary diversity scores may be an interesting assessment tool in the public health context, despite some limitations, because it is a clear, objective instrument (since it is not susceptible to the subjectivity of the classification of foods and groups), and it is easy to use for professionals. The concept of variety is also easily understood by the population, and the food groups are wide, so it can be a good tool for counseling older adults, considering, of course, joint guidance on how to select foods from each group according to the degree of processing.

## Figures and Tables

**Table 1 foods-13-03449-t001:** Prevalence (%) of consumption of food group indicators (G) according to sex and age group. Brazil, 2018–2019 (n = 581).

	Total n (%)	Sex	
		Male %	Female %
G1: Grains, white roots and tubers, and plantains	580 (99.83)	100.00	99.75
G2: Pulses (beans, peas, and lentils)	238 (40.96)	63.48	31.02 *
G3: Nuts and seeds	33 (5.68)	3.93	6.45
G4: Dairy	444 (76.42)	74.16	77.42
G5: Meat, poultry, and fish	500 (86.06)	87.08	85.61
G6: Eggs	153 (26.33)	26.97	26.05
G7: Dark green leafy vegetables	103 (17.73)	12.36	20.10 *
G8: Other vitamin-A-rich fruits and vegetables	100 (17.21)	12.92	19.11 *
G9: Other vegetables	349 (60.07)	55.62	52.03
G10: Other fruits	445 (76.59)	68.54	80.15 *
	Total n (%)	Age group	
		60–69 years-old %	≥70 years-old %
G1: Grains, white roots and tubers, and plantains	580 (99.83)	100.00	99.64
G2: Pulses (beans, peas, and lentils)	238 (40.96)	39.07	43.01
G3: Nuts and seeds	33 (5.68)	5.96	5.38
G4: Dairy	444 (76.42)	72.85	80.29 *
G5: Meat, poultry, and fish	500 (86.06)	86.75	85.30
G6: Eggs	153 (26.33)	30.13	22.22 *
G7: Dark green leafy vegetables	103 (17.73)	18.21	17.20
G8: Other vitamin-A-rich fruits and vegetables	100 (17.21)	14.24	20.43
G9: Other vegetables	349 (60.07)	61.59	58.42
G10: Other fruits	445 (76.59)	77.48	75.63

Note: * *p* < 0.05; x^2^ test.

**Table 2 foods-13-03449-t002:** Mean (standard deviation—SD) of the dietary diversity score (DDS) and prevalence (%) of minimum dietary diversity (MDD—consuming five or more food groups) according to main sociodemographic characteristics. Brazil, 2018–2019 (n = 581).

	n	DDS		MDD ^c^	
		Mean (SD)	*p*	%	*p*
Sex			0.827 ^a^		0.715
Male	178	5.05 (1.28)		68.55	
Female	403	5.08 (1.36)		67.00	
Age group			0.912 ^a^		0.566
60–69 years old	302	5.06 (1.33)		68.54	
≥70 years	279	5.07 (1.35)		66.31	
Ethnicity			0.257 ^a^		0.385
Caucasians	321	5.13 (1.36)		69.16	
African Americans, others	257	5.00 (1.31)		65.76	
Living arrangement			0.131		0.164
Living with other people	483	5.03 (1.35)		66.25	
Living alone	98	5.26 (1.29)		73.47	
Family monthly income					0.028
>1 minimum wages	190	4.73 (1.30)		60.00	
1–2 minimum wages	245	5.17 (1.35)	0.001 ^b^	71.02	
≤2 minimum wages	146	5.34 (1.30)	<0.001 ^b^	71.23	
Schooling (years)			0.003 ^a^		0.030
0–4	295	4.90 (1.32)		63.05	
5 and more	281	5.23 (1.35)		71.53	

Note: minimum wage: considered the minimum monthly income received at the time of interview [2018 = BRL 954.00 (USD 261.1); 2019 = BRL 998.00 (USD 257.2)]. ^a^
*p*-value for Student’s *t*-test; ^b^
*p*-value for linear regression; ^c^
*p*-value for x^2^ test.

**Table 3 foods-13-03449-t003:** Mean (standard deviation—SD) of the dietary diversity score (DDS) and prevalence (%) of minimum dietary diversity (MDD—consuming five or more food groups) according to main health conditions and lifestyle characteristics. Brazil, 2018–2019 (n = 581).

	n	DDS ^a^		MDD ^c^	
		Mean (SD)	*p*	Mean (SD)	*p*
Hypertension			0.292		0.376
No	224	5.14 (1.32)		69.64	
Yes	357	5.02 (1.35)		66.11	
Diabetes mellitus			0.896		0.552
No	422	5.07 (1.36)		66.82	
Yes	157	5.08 (1.31)		69.43	
Cardiovascular diseases			0.974		0.727
No	441	5.07 (1.32)		68.03	
Yes	134	5.07 (1.40)		66.42	
Cancer			0.009 ^a^		0.007
No	538	5.03 (1.34)		65.99	
Yes	43	5.58 (1.18)		86.05	
Anorexia of aging			0.001 ^a^		0.033
No	432	5.17 (1.34)		69.91	
Yes	149	4.77 (1.31)		60.40	
Cognitive status			<0.001 ^a^		<0.001
Preserved	518	5.15 (1.32)		69.88	
Declined	63	4.43 (1.36)		47.62	
Physical activity					<0.001
Active	262	5.32 (1.26)		75.19	
Insufficiently active	112	5.18 (1.24)	0.325 ^b^	69.64	
Sedentary	207	4.69 (1.41)	<0.001 ^b^	56.52	
Smoking			0.005 ^a^		0.003
Non-smokers	542	5.11 (1.31)		69.00	
Smokers	39	4.49 (1.55)		46.15	
Alcohol intake					0.108
Never/rarely	442	5.02 (1.36)		65.61	
1 to 4 times a month	91	5.44 (1.13)	0.006 ^b^	76.92	
At least 1 time a week	47	4.79 (1.37)	0.258 ^b^	65.96	

Note: ^a^ *p*-value for Student’s *t*-test; ^b^ *p*-value for linear regression; ^c^ *p*-value for x^2^ test.

**Table 4 foods-13-03449-t004:** Factors associated with diet diversity score (DDS). Brazil, 2018–2019 (n = 581).

	Model 1		Model 2	
	ꞵ	IC (95%)	ꞵ	IC (95%)
Family monthly income				
>1 minimum wages (ref.)	-	-	-	-
1–2 minimum wages	0.42 **	0.16; 0.67	0.37 **	0.13; 0.62
≤2 minimum wages	0.54 **	0.25; 0.83	0.43 **	0.14; 0.72
Schooling (up to 4 years)	−0.25 *	−0.47; −0.03	−0.15	−0.37; 0.06
Social support score	0.01 **	0.00; 0.01	0.00	0.00; 0.01
Cancer			0.48 *	0.08; 0.88
Anorexia of aging			−0.30 *	−0.54; −0.05
Cognitive decline			−0.46 **	−0.81; −0.11
Physical activity				
Active (ref.)			-	-
Insufficiently active			−0.11	−0.40; 0.17
Sedentary			−0.45 **	−0.69; −0.21
Alcohol intake				
Never/rarely (ref.)			-	-
1 to 4 times a month			0.30 *	0.00; 0.59
At least 1 time a week			−0.37	−0.76; 0.02

Note: model 1: adjusted by sociodemographic block; model 2: model 1 + variables of health conditions and lifestyle indicators. 95% CI = 95% confidence interval. * *p* < 0.05 ** *p* < 0.01.

## Data Availability

The datasets analyzed during the current study are not publicly available but are available from the corresponding author on reasonable request.

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
