# Peer review of "Factors Associated with Dietary Diversity in Community-Dwelling Brazilian Older Adults"

_foods, 2024, doi:10.3390/foods13213449_

Round 1
Reviewer 1 Report
Comments and Suggestions for Authors
The authors' research results might be applied to guide the specialized institutions responsible for elder care. Some recommendations for the researchers are the following:
· Motivate the connections made to the Minimum Dietary Diversity for Women scale since the system considers females at reproductive age. Your analysis and possible advice are for elderly persons.
· The Introduction section needs additional information to underline the state of the art in the field and properly sustain the necessity of such a study.
· Review the reference system as consecutive numbering as well as format. The Introduction section indicates 10 bibliographic positions, and the Material and Methods section continues with 53, 56.
· Statistical analysis was used. To improve the significance of the results, consider including the hypothesis from which you started the research and whether they were confirmed or not.
· As a table, include the socio-demographic characteristics of the respondents you used to make the correlations (gender, family income, health limitations, number of respondents from each city, etc.).
· Specify the number of socially assisted persons from each institution who might have corresponded to all your selection criteria and the number of those who agree to be included in the study to highlight the study's representativeness.
After major changes, the paper could be considered for publication. The authors must revise and resubmit it with suggested modifications specified in the reviewer's comments.
Comments on the Quality of English LanguageMinor editing of English language required.
Author Response
Comment 1: The authors' research results might be applied to guide the specialized institutions responsible for elder care. Some recommendations for the researchers are the following.
Response 1: We thank you for your time and careful reading. We only would like to reinforce that the research was conducted with community-dwelling older adults, not people living in nursing homes or other specialized institutions for elderly.
Comment 2: Motivate the connections made to the Minimum Dietary Diversity for Women scale since the system considers females at reproductive age. Your analysis and possible advice are for elderly persons.
Response 2: The MDD-W is a food group diversity indicator that, despite being proposed as a proxy indicator to reflect the micronutrient adequacy of women’s diets, had been used with several other populational groups. We opted to use this tool because older adults are also nutritionally vulnerable and nutrient requirements are higher for older than for younger adults. Also, it has several other advantages, and we added one paragraph in Introduction explaining this indicator more broadly, its strengths and studies that used it in other age groups (lines 105-121). Besides, this fact is also presented as limitation of the study in the end of the discussion section.
Comment 3: The Introduction section needs additional information to underline the state of the art in the field and properly sustain the necessity of such a study.
Response 3: We agree that the introduction does not contain enough information to justify the study, so we reformulated this section, strengthen the literature review and the rationale of the study. We hope that is more adequate in this new version.
Comment 4: Review the reference system as consecutive numbering as well as format. The Introduction section indicates 10 bibliographic positions, and the Material and Methods section continues with 53, 56.
Response 4: Thank you for pointing this out. We corrected the sequence along the text.
Comment 5: Statistical analysis was used. To improve the significance of the results, consider including the hypothesis from which you started the research and whether they were confirmed or not.
Response 5: We agree that this is an important issue. So, we added a general hypothesis (lines 215-219), and we hope it is enough to justify the option for the tests. We also made some minor changes to clarify the text, as suggested by reviewer 2, so statistical analysis is better understood. Please feel free to suggest more changes if is still not clear enough.
Comment 6: As a table, include the socio-demographic characteristics of the respondents you used to make the correlations (gender, family income, health limitations, number of respondents from each city, etc.).
Response 6: Thank you for the suggestions. However, we consider that this inclusion is not possible because these data have already been presented as tables in previous publications and could be considered plagiarism (please see references 16 and 17). Besides, the number of respondents in each category is already presented in tables 2 and 3. So, we added the main descriptive variables in the first paragraph of the results section (lines 227-231), and we hope this is enough to characterize the sample. We also included the number of respondents from each city in methods section (lines 146-147).
Comment 7: Specify the number of socially assisted persons from each institution who might have corresponded to all your selection criteria and the number of those who agree to be included in the study to highlight the study's representativeness.
Response 7: We would like to clarify that the research was conducted with community-dwelling older adults, not people living in nursing homes or other specialized institutions for elderly. We added some concise description to clarify the basic information about the sampling and recruiting methods (lines 133-145), as suggested by reviewer 2, and we hope is clear enough in this new version.
Comment 8: After major changes, the paper could be considered for publication. The authors must revise and resubmit it with suggested modifications specified in the reviewer's comments.
Response 8: We thank you for your time and careful reading. We tried to address all concerns and we hope the paper has improved in this new version.
Comment 9: Comments on the Quality of English Language: Minor editing of English language required.
Response 9: Thank you for pointing this out. We conducted a careful English review. We hope is more adequate in this new version.
Reviewer 2 Report
Comments and Suggestions for Authors
Dear Authors,
The abstract is very well written, it is clearly structured and gives a complete picture of the content of the full article.
If more recent data are available for the following paragraph (Ac-38 cording to the World Health Organization (WHO), in 2020, the global population aged 60 39 years and over will be just over 1 billion people, representing 13.5% of the world popula-40 tion, with two thirds of older people living in low- and middle-income countries (LMIC) ), the authors are kindly requested to update them. It is already 2024, so more recent population data may be available.
The introduction, which currently also contains the literature study, is too short and does not contain enough information. I recommend separating the introduction from the literature study. The introduction should contain the research problem, the identified gap, the purpose of the research and finally the way the article is structured. Then, the literature review should focus on reviewing the literature in the field, citing the most recent literature sources.
The shortcoming of this paragraph "This is a cross-sectional study which aimed to evaluate data on food consumption 78 and micronutrient deficiency in elderly people in three cities (Campinas, Limeira and Pi-79 racicaba) in the State of Sao Paulo, Brazil. A convenience sample was selected in primary 80 care units recommended by the Health Departments of each municipality. At each health 81 unit, users who met the inclusion criteria (aged 60 years or older, living in one of the par-82 ticipating municipalities, being registered in the Family Health Strategy (FHS)), were in-83 vited to participate in the research. More detail about sampling and recruitment were 84 published previously "is that it does not provide sufficient details about the process of sample selection and recruitment in the study, instead referring to another publication for this information. Even if the data have been previously published, for clarity and completeness, it is advisable that each article include a concise description of the methodology so that readers do not have to search for other sources to fully understand the study presented. This approach is not appropriate because each article should stand alone and contain all the essential information to be fully understood.
The questionnaire should be presented as an annex or additional material to the article.
The presentation of the statistical analysis is generally clear and well structured, but could benefit from some improvements to ensure greater clarity and coherence. Here are some comments and suggestions for improving the section: Use consistent and precise language, avoiding unnecessary repetition. Explanations of statistical methods should be presented in a way that is easy for non-statisticians to understand; explain in more detail why certain statistical tests were used (e.g., why the Shapiro-Wilk test was used and the implications of normality for the choice of subsequent tests).
The discussion section of the study provides a comprehensive analysis of the findings and places them in the broader context of the existing literature. Several areas could be improved to enhance clarity, coherence and depth. The discussion is long and covers a wide range of issues. It might benefit from being divided into more clearly defined subsections. This would make it easier for the reader to follow the narrative. Some paragraphs are very dense with information. Consider breaking them up into shorter, more focused paragraphs. There is some repetition, particularly when discussing gender differences in dietary habits. Streamlining these sections could improve readability. The discussion of the Brazilian National Dietary Survey and other studies could be more concise to avoid redundancy. While the section provides a lot of comparative data, the analysis of why certain trends are observed could be deepened. For example, the reasons for gender differences in food consumption or the impact of socio-economic status could be explored in more detail. The possible reasons for the higher consumption of dairy products by older people are touched upon but not fully explored.The discussion references many studies, but sometimes the integration of these studies into the overall argument feels forced. Ensure that each referenced study directly supports or contrasts your findings in a meaningful way. Some studies are mentioned without a clear connection to the main argument. Make sure that each reference clearly contributes to the understanding of your findings.
In the concluding part, the authors should better explain what are the practical, managerial implications of the study and to what extent its results contribute to the advancement of the literature.
Author Response
Comment 1: Dear Authors,
The abstract is very well written, it is clearly structured and gives a complete picture of the content of the full article.
Response 1: We thank you for your time and careful reading. We tried to address all concerns and we hope the paper has improved in this new version.
Comment 2: If more recent data are available for the following paragraph (According to the World Health Organization (WHO), in 2020, the global population aged 60 years and over will be just over 1 billion people, representing 13.5% of the world population, with two thirds of older people living in low- and middle-income countries (LMIC)), the authors are kindly requested to update them. It is already 2024, so more recent population data may be available.
Response 2: Thank you for pointing this out. We added an updated reference to the paragraph.
Comment 3: The introduction, which currently also contains the literature study, is too short and does not contain enough information. I recommend separating the introduction from the literature study. The introduction should contain the research problem, the identified gap, the purpose of the research and finally the way the article is structured. Then, the literature review should focus on reviewing the literature in the field, citing the most recent literature sources.
Response 3: We agree that the introduction does not contain enough information to justify the study, so we strengthen the literature review and the rationale of the study. We hope that is more adequate in this new version.
Comment 4: The shortcoming of this paragraph "This is a cross-sectional study which aimed to evaluate data on food consumption and micronutrient deficiency in elderly people in three cities (Campinas, Limeira and Piracicaba) in the State of Sao Paulo, Brazil. A convenience sample was selected in primary care units recommended by the Health Departments of each municipality. At each health unit, users who met the inclusion criteria (aged 60 years or older, living in one of the participating municipalities, being registered in the Family Health Strategy (FHS)), were invited to participate in the research. More detail about sampling and recruitment were published previously "is that it does not provide sufficient details about the process of sample selection and recruitment in the study, instead referring to another publication for this information. Even if the data have been previously published, for clarity and completeness, it is advisable that each article include a concise description of the methodology so that readers do not have to search for other sources to fully understand the study presented. This approach is not appropriate because each article should stand alone and contain all the essential information to be fully understood.
Response 4: Thank you for point this out. We added some concise description to clarify the basic information about the sampling and recruiting methods (lines 133-145), and we hope is clear enough in this new version.
Comment 5: The questionnaire should be presented as an annex or additional material to the article.
Response 5: Thank you for your suggestion. We did not present the questionnaire because it is in Brazilian Portuguese, but if the reviewers and editors consider that is important, we can present as supplemental material. We indicated this in lines 174-175).
Comment 6: The presentation of the statistical analysis is generally clear and well structured, but could benefit from some improvements to ensure greater clarity and coherence. Here are some comments and suggestions for improving the section: Use consistent and precise language, avoiding unnecessary repetition. Explanations of statistical methods should be presented in a way that is easy for non-statisticians to understand; explain in more detail why certain statistical tests were used (e.g., why the Shapiro-Wilk test was used and the implications of normality for the choice of subsequent tests).
Response 6: Thank you for the suggestions. Nevertheless, we did not find many points that could be clarified or removed because of repetition, so we made minor changes and included the hypothesis, as suggested by reviewer 1. For example, Shapiro Wilk test is the most common test to verify normality, so we do not know exactly how to improve the sentence. Please feel free to suggest more changes if is still not clear enough.
Comment 7: The discussion section of the study provides a comprehensive analysis of the findings and places them in the broader context of the existing literature. Several areas could be improved to enhance clarity, coherence and depth. The discussion is long and covers a wide range of issues. It might benefit from being divided into more clearly defined subsections. This would make it easier for the reader to follow the narrative. Some paragraphs are very dense with information. Consider breaking them up into shorter, more focused paragraphs. There is some repetition, particularly when discussing gender differences in dietary habits. Streamlining these sections could improve readability. The discussion of the Brazilian National Dietary Survey and other studies could be more concise to avoid redundancy. While the section provides a lot of comparative data, the analysis of why certain trends are observed could be deepened. For example, the reasons for gender differences in food consumption or the impact of socio-economic status could be explored in more detail. The possible reasons for the higher consumption of dairy products by older people are touched upon but not fully explored. The discussion references many studies, but sometimes the integration of these studies into the overall argument feels forced. Ensure that each referenced study directly supports or contrasts your findings in a meaningful way. Some studies are mentioned without a clear connection to the main argument. Make sure that each reference clearly contributes to the understanding of your findings.
Response 7: Thank you for your valuable suggestions. We agree that the discussion is too dense, so we added 3 subsections: Prevalence of consumption by food groups; Factors associated to dietary diversity; and Strengths and limitations. We hope this improved readability. We also excluded some references that seemed to be out of the scope or excessive, as well as some phrases or paragraphs. We hope the discussion has improved in this new version. About the possible reasons for higher consumption of dairy products by older people are not possible to discuss any further, because of the cross-sectional nature of the study. We only raised the hypothesis that the older participants in the present study have already received guidance from health professionals about the importance of dairy products in bone health, but we believe we cannot extrapolate anymore than that. If the reviewer prefers to indicate some additional literature or hypothesis to understand that phenomena, we are more than happy to incorporate them.
Comment 8: In the concluding part, the authors should better explain what are the practical, managerial implications of the study and to what extent its results contribute to the advancement of the literature.
Response 8: Thank you for your suggestion. We rewrote the conclusion, indicating the practical implications and how our findings advance in literature. We hope is clear enough in this new version.
Reviewer 3 Report
Comments and Suggestions for Authors
Dear author,
Thank you to give the opportunity to review your article in general it is good. But I have some suggestions to improve it.
Introduction
Line 73 a reference is needed.
Material and methods
Line 85, the references go from number 10 to 53, to be reordered in the rest of the manuscript.
Results
Should be described the characteristics of the population enrolled in the study.
Best regards,
Author Response
Comment 1: Dear author, Thank you to give the opportunity to review your article in general it is good. But I have some suggestions to improve it.
Response 1: We thank you for your time and careful reading. We tried to address all concerns and we hope the paper has improved in this new version.
Comment 2: Introduction: Line 73 a reference is needed.
Response 2: Thank you for pointing this out. We added an updated reference to the paragraph.
Comment 3: Material and methods: Line 85, the references go from number 10 to 53, to be reordered in the rest of the manuscript.
Response 3: Thank you for pointing this out. We corrected the sequence along the text.
Comment 4: Results: Should be described the characteristics of the population enrolled in the study.
Response 4: Thank you for the suggestion. The number of respondents in each category is already presented in tables 2 and 3. So, we added the main descriptive variables in the first paragraph of the results section (lines 227-231), and we hope this is enough to characterize the sample. We also included the number of respondents from each city in methods section (lines 146-147).